# Prostate-specific antigen dynamics after neoadjuvant androgen-deprivation therapy and carbon ion radiotherapy for prostate cancer

**Yosuke Takakusagi**[1]*, **Takahiro Oike**[2], **Kio Kano**[1], **Wataru Anno**[1], **Keisuke Tsuchida**[1], **Nobutaka Mizoguchi**[1], **Itsuko Serizawa**[1], **Daisaku Yoshida**[1], **Hiroyuki Katoh**[1], **Tadashi Kamada**[1]

1 Department of Radiation Oncology, Kanagawa Cancer Center, Yokohama, Kanagawa, Japan,
2 Department of Radiation Oncology, Gunma University Graduate School of Medicine, Maebashi, Gunma, Japan

* y-takakusagi@kcch.jp

**Data Availability Statement:** All relevant data are within the paper.

## Abstract

### Background

This study aimed to explain the dynamics of prostate-specific antigen (PSA) levels in patients with prostate cancer who were treated with carbon ion radiotherapy (CIRT) and neoadjuvant androgen-deprivation therapy (ADT).

### Methods

Eighty-five patients with intermediate-risk prostate cancer who received CIRT and neoadjuvant ADT from December 2015 to December 2017 were analyzed in the present study. The total dose of CIRT was set at 51.6 Gy (relative biological effectiveness) delivered in 12 fractions over 3 weeks. The PSA bounce was defined as a ≥0.4 ng/ml increase of PSA levels from the nadir, followed by any decrease. PSA failure was defined using the Phoenix criteria.

### Results

The median patient age was 68 (range, 48–81) years. The median follow-up duration was 33 (range, 20–48) months. The clinical T stage was T1c, T2a, and T2b in 27, 44, and 14 patients, respectively. The Gleason score was 6 in 3 patients and 7 in 82 patients. The median pretreatment PSA level was 7.37 (range, 3.33–19.0) ng/ml. All patients received neoadjuvant ADT for a median of 6 (range, 2–117) months. PSA bounces were observed in 39 patients (45.9%), occurring a median of 12 (range, 6–30) months after CIRT. PSA failure was observed in eight patients (9.4%), occurring a median of 21 (range, 15–33) months after CIRT. The 3-year PSA failure-free survival rate was 88.5%. No clinical recurrence was observed during the follow-up period. Younger age and lower T stage were significant predictors of PSA bounce. Younger age was a significant predictor of PSA failure.

**Funding:** The authors received no specific funding for this work.

**Competing interests:** The authors have declared that no competing interests exist.

**Abbreviations:** ADT, androgen-deprivation therapy; AUC, area under the curve; CIRT, carbon ion radiotherapy; CT, computed tomography; CTV, clinical target volume; HDR-BT, high-dose-rate brachytherapy; IMRT, intensity-modulated radiotherapy; i-ROCK, ion-beam Radiation Oncology Center in Kanagawa; LDR-BT, low-dose-rate brachytherapy; MRI, magnetic resonance imaging; PSA, prostate-specific antigen; PTV, planning target volume; RBE, relative biological effectiveness; ROC, receiver operating characteristic; SRT, stereotactic radiation therapy.

## Conclusions

In this study, we identified the significant predictors of the occurrence of PSA bounce and failure. Further follow-up is needed to reveal the clinical significance of PSA dynamics.

## Background

Among cancers, prostate cancer ranks second globally in morbidity and fifth in mortality [1]. Radiotherapy is one of the definitive treatments for localized or locally advanced prostate cancer. The number of patients treated with radiotherapy for prostate cancer has been increasing in Japan according to a structural survey conducted by the Japanese Society for Therapeutic Radiology and Oncology [2]. Brachytherapy, intensity-modulated radiotherapy (IMRT), and particle beam radiotherapy are the radiotherapy modalities used for patients with prostate cancer [3–6].

The first carbon ion radiotherapy (CIRT) clinical trial for prostate cancer was initiated in 1995 at the National Institute of Radiological Sciences (Chiba, Japan) [7]. CIRT offers biological and physical advantages over conventional photon radiotherapy with X-rays. Regarding the biological aspect, carbon ion beams have an estimated 2–3-fold higher relative biological effectiveness (RBE) than X-rays [8, 9]. In terms of the physical aspect, a more conformal dose distribution can be delivered via CIRT based on the ability of accelerated carbon ions to release a maximal amount of energy at the end of their track, resulting in a Bragg peak [10]. These features have led to favorable clinical outcomes for CIRT in prostate cancer [6, 11]. In the ion-beam Radiation Oncology Center in Kanagawa (i-ROCK), similar to previous studies of CIRT for prostate cancer, favorable clinical outcomes were achieved for prostate cancers treated with CIRT [12].

Serum prostate-specific antigen (PSA) is a sensitive marker of treatment outcomes for prostate cancer [13]. Fluctuation of PSA levels is often observed after radiotherapy without any clinical recurrence [14–16]. Such benign PSA fluctuation, which was first reported in 1997, is known as the PSA bounce [17]. PSA bounces can be disconcerting for patients and physicians [18], and they may lead to unnecessary salvage treatment for cases that meet the definition of PSA failure. Therefore, accurate clinical interpretation of PSA dynamics after radiotherapy for prostate cancer is necessary to avoid patient anxiety or a false-positive diagnosis of relapse, which can instigate unnecessary treatment [19].

PSA bounces have been observed after various radiotherapy modalities for prostate cancer, such as low-dose-rate brachytherapy (LDR-BT), high-dose-rate brachytherapy (HDR-BT), IMRT, and stereotactic radiotherapy (SRT). However, only one study has reported PSA dynamics after CIRT [20]. In that study, although PSA dynamics after CIRT alone was revealed, that after CIRT using androgen-deprivation therapy (ADT) was not investigated. Thus, the present study aimed to explain the dynamics of PSA in patients with prostate cancer who were treated with CIRT and ADT.

## Materials and methods

The study was approved by the institutional review board of Kanagawa Cancer Center (approval number: 2019–171). Written informed consent was obtained from all patients. Clinical data obtained between December 2015 and December 2019 was accessed in this study. The source of medical records used in this work was Kanagawa Cancer Center.

## Patients

In total, the cases of 85 consecutive patients with intermediate-risk prostate cancer who received CIRT at i-ROCK between December 2015 and December 2017 were analyzed in the present study. The patients were classified using the D'Amico risk group classification [21]. The clinical stage was determined using computed tomography (CT), magnetic resonance imaging (MRI), bone scintigraphy, and other diagnostic images. The eligibility criteria for this study were as follows: (i) histological diagnosis of prostate adenocarcinoma, (ii) cT1cN0M0 to T2bN0M0 according to the 7th UICC classification, (iii) performance status of 0–2, (iv) age of 20 years or older, (v) no previous treatment for prostate cancer excluding ADT, and (vi) followed up at least 1 year post-CIRT.

## CIRT

All patients were treated at i-ROCK in Japan. The first clinical treatment for prostate cancer at the i-ROCK was performed in 2015 [22].

Patients were placed in the supine position on a vacuum mattress (BlueBAG: Elekta AB, Stockholm, Sweden) and immobilized using thermoplastic shells (Shellfitter: Kuraray, Tokyo, Japan). Enema was used before CT for CIRT planning. The rectum was emptied as much as possible using a laxative and an antiflatulent before each session, and enema was performed if the patient did not defecate within 24 h of treatment. The patients urinated and drank water 60 min before CT. A set of CT images with 2-mm-thick slices was taken for treatment planning.

Contouring of the target volumes and normal tissues was performed using MIM maestro software version 5.6 (MIM Software Inc., Cleveland, OH, USA). Dose calculation and optimization were performed using the Monaco version 5.20 system (Elekta AB).

The prostate volume was measured via CT imaging. The gross tumor volume was not defined. The clinical target volume (CTV) included the entire prostate and proximal seminal vesicles. Planning target volume (PTV) 1 was created by adding anterior and lateral margins of 10 mm and a posterior margin of 5 mm to the CTV. Boost therapy was performed using PTV2, in which the posterior edge was set in front of the anterior wall of the rectum to reduce the rectal dose in the ninth course of treatment [23, 24]. The rectum was delineated as the organ at risk from 10 mm above the upper margin of the PTV to 10 mm below the lower margin of the PTV.

The total dose was set at 51.6 Gy (RBE). After the first eight fractions were delivered using PTV1, boost therapy was performed using PTV2 in the latter four fractions. The PTV was covered by ≥95% of the prescribed dose, and the maximum PTV dose was limited to <105% of the prescribed dose. The dose constraint for the rectum aimed at V80% < 10 ml.

CIRT was administered once daily for 4 days a week for 3 weeks. All patients were treated using the spot scanning method. CIRT was performed from both the right and left sides of the patient. One port was used for each treatment session. Verification of the patient position was performed using in-room CT during the first, fifth, and ninth treatment sessions. In each treatment session, a computer-aided online positioning system was employed to verify the positioning accuracy to less than 1 mm.

## ADT

Urologists administered ADT. Neoadjuvant ADT was administered for 4–8 months through the end of CIRT [6]. ADT was performed via combined androgen blockade with an antiandrogen plus medical castration in principle. We performed a representative ADT using a combination of bicalutamide and leuprorelin acetate.

### Follow-up

A urologist and a radiation oncologist conducted patient follow-up at 3-month intervals for the first 3 years after CIRT and at 6-month intervals thereafter. PSA was measured at each follow-up visit. In the present study, the PSA bounce was defined as a PSA increase of at least 0.4 ng/ml from the nadir PSA level, followed by any decrease [25, 26]. PSA failure was defined using the Phoenix definition, namely, the nadir PSA level plus 2 ng/ml [27]. The follow-up period and the time to the event were calculated from the start of CIRT to the date of the event.

### Statistical analysis

Statistical analysis was performed using STATA software (version 13.1, TX, USA). The correlation of clinical variables with PSA dynamics was assessed via the Cox regression analyses. Prognostic factors, for which $p$ value was calculated as <0.10, were evaluated using the multivariate stepwise Cox regression model [28]. Comparative analyses for continuous variables, such as PSA level and age of the two groups, were examined using the Mann–Whitney U test. Non-parametric receiver operating characteristic (ROC) curves were generated and Youden index (J = max [sensitivity + specificity– 1]) was used to determine the optimal cut-off values [29]. Comparative analyses for categorical variables of the two groups were examined using the chi-squared test. A $p$ value of <0.05 was considered significant. The PSA failure-free survival rate was estimated using the Kaplan–Meier method.

## Results

### Patient characteristics

Patient characteristics are summarized in Table 1. The median patient age was 68 (range, 48–81) years. The median follow-up duration was 33.1 (range, 20.1–48.3) months. All patients completed CIRT on schedule. Neoadjuvant ADT was administered to all patients, and the median duration of ADT was 6.2 (range, 2.3–116.9) months. The patient who received ADT for 116.9 months, which is an outlier, was treated via combined androgen blockade in the first year. PSA elevation was observed 4 years after initial ADT, and the patient was treated by antiandrogen alone irregularly when PSA elevation was observed. Four months of combined androgen blockage was performed in this patient before CIRT. Pre-CIRT PSA levels were measured a median of 15 (range, 0–40) days before the start of CIRT.

### PSA dynamics

PSA dynamics for all patients is presented in Fig 1(a). The average PSA dynamics based on the presence or absence of the PSA bounce is presented in Fig 1(b). The average PSA level in the PSA bounce group was significantly higher than that in the PSA bounce-free group beyond 3 months after CIRT ($p < 0.05$). The average PSA dynamics for the presence or absence of PSA failure is presented in Fig 1(c). The average PSA level in the PSA failure group was significantly higher than that in the PSA failure-free group before CIRT and at all time points between 6 and 36 months after CIRT, excluding 30 months ($p < 0.05$).

PSA bounces were observed in 39 patients (45.9%) a median of 12 (range, 6–30) months after CIRT. Predictive significance of clinical variables for the occurrence of PSA bounces was assessed via Cox regression analysis (Table 2). In the univariate analysis, younger age, lower T stage, and higher PSA nadir were significantly associated with the occurrence of a PSA bounce ($p < 0.001$, 0.015, and 0.029, respectively). The median ages of patients with and without PSA bounces were 68 (range, 48–79) and 70 (range, 55–81) years, respectively ($p = 0.001$). The T stage in the PSA bounce group was T1c, T2a, and T2b in 16 (41.0%), 20 (51.3%), and 3 (7.7%)

**Table 1. Patient characteristics (n = 85).**

| Characteristics | n (%) |
|---|---|
| Follow-up duration, months, median (range) | 33.1 (20.1–48.3) |
| Age, years, median (range) | 68 (48–81) |
| T stage | |
| 1c | 27 (31.8%) |
| 2a | 44 (51.8%) |
| 2b | 14 (16.5%) |
| Pretreatment PSA, ng/ml, median (range) | 7.37 (3.33–19.0) |
| < 10 | 62 72.9%) |
| 10 ≤ 20 | 23 (27.1%) |
| Gleason score | |
| 6 | 3 (3.5%) |
| 7 | 82 (96.5%) |
| D'Amico classification | |
| intermediate | 85 (100.0%) |
| ADT | |
| neoadjuvant | 85 (100.0%) |
| duration, month, median (range) | 6.2 (2.3–116.9) |
| Prostate volume, cc, median (range) | 26.9 (11.9–88.2) |
| pre-CIRT PSA, ng/ml, median (range) | 0.31 (0.01–3.28) |
| Time to nadir PSA, month, median (range) | 3 (3–24) |

PSA: Prostate specific antigen, ADT: Androgen deprivation therapy.

patients, respectively, versus 11 (23.9%), 24 (52.2%), and 11 (23.9%) patients, respectively, in the PSA bounce-free group ($p$ = 0.027). The median PSA nadir of patients with and without PSA bounces were 0.014 ng/ml (range, 0–2.183 ng/ml) and 0 ng/ml (range, 0–0.201 ng/ml), respectively ($p$ = 0.002). ADT duration was not correlated with the occurrence of PSA bounce ($p$ = 0.731). In the multivariate analyses, younger age and lower T stage were significantly associated with the occurrence of a PSA bounce ($p$ < 0.001 and 0.049, respectively). The ROC curve analysis calculated the area under the ROC curve (AUC) as 0.705 and determined a cut-off value of 68 years, at which the sensitivity and specificity were measured to be 76.1 and 61.5%, respectively (Fig 2a).

PSA failure was observed in eight patients (9.4%). The characteristics of eight patients with PSA failure are summarized in Table 3. Of these eight patients with PSA failure, the PSA level decreased in seven patients without any treatment such as ADT in a median of 3 (range, 3–15) months after PSA failure. No salvage treatments were performed in these seven patients in the follow-up period. The remaining patient received ADT immediately after the occurrence of PSA failure without radiological confirmation of clinical recurrence. As shown in Fig 3, the 3-year PSA failure-free rate was 88.5%. PSA failure occurred in a median of 21 (range, 15–33) months after CIRT. Clinical recurrence was not detected by CT, MRI, and bone scintigraphy.

The predictive significance of clinical variables for the occurrence of PSA failure was assessed via the Cox regression analyses (Table 4). In the univariate analysis, younger age, higher pre-CIRT PSA levels, and higher PSA nadir were significantly associated with the occurrence of PSA failure ($p$ = 0.002, 0.007, and 0.003, respectively). The median ages of patients with and without PSA failure were 61 (range, 50–68) and 69 (range, 48–81) years, respectively ($p$ = 0.001). The median pre-CIRT PSA levels of patients with and without PSA

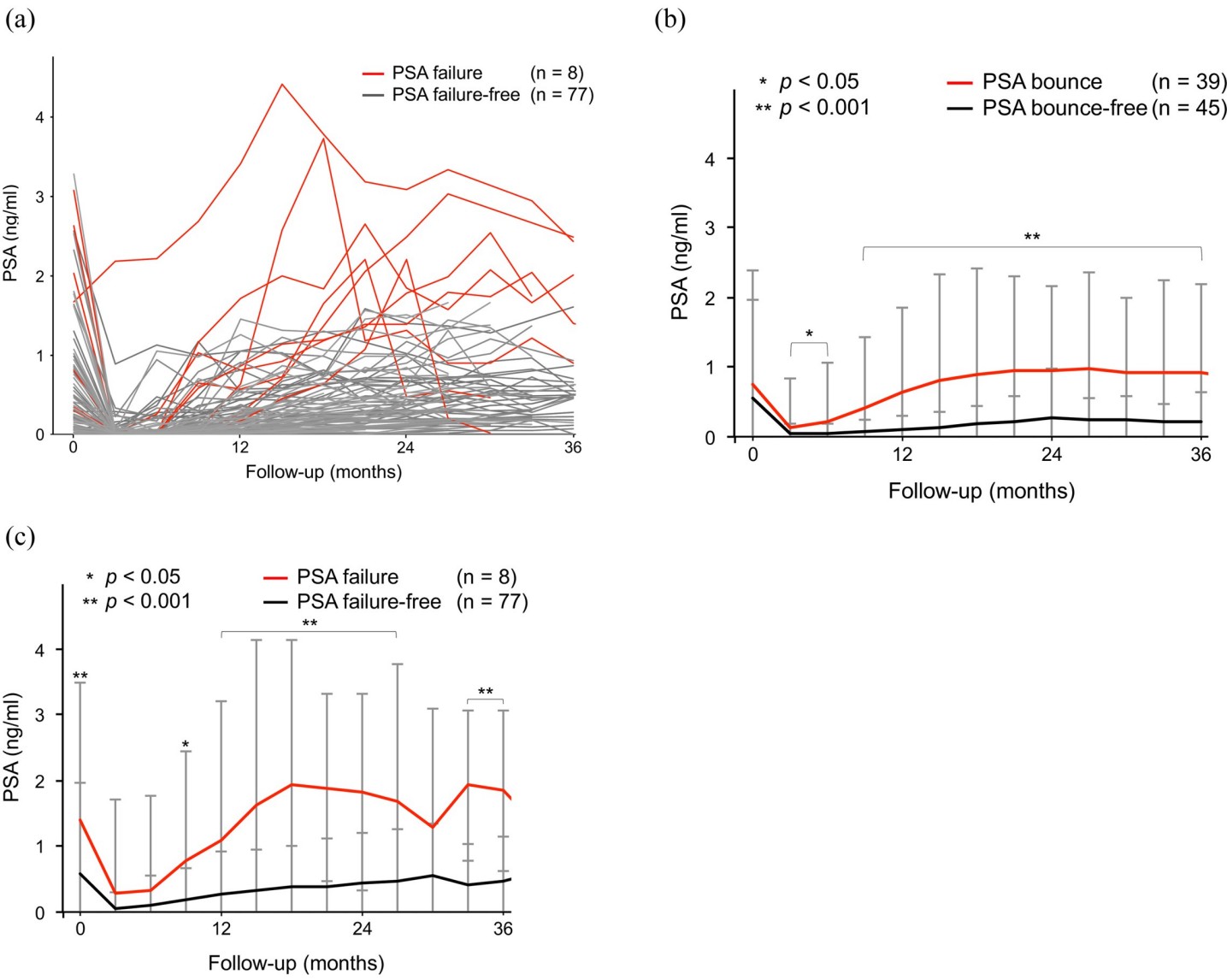

**Fig 1. (a).** Prostate-specific antigen (PSA) dynamics after carbon ion radiotherapy (CIRT) for all patients. The PSA dynamics in patients with biochemical relapse is indicated by the red line. No clinical recurrence was observed. **(b).** The average PSA dynamics in patients with or without PSA bounces. **(c).** The average PSA dynamics in patients with or without PSA failure.

failure were 1.24 (range, 0.30–3.97) and 0.25 (range, 0.01–3.28) ng/ml, respectively ($p = 0.009$). The median PSA nadir of patients with and without PSA bounces were 0.014 ng/ml (range, 0–2.183 ng/ml) and 0 ng/ml (range, 0–0.201 ng/ml), respectively ($p = 0.002$). ADT duration was not correlated with the occurrence of PSA failure ($p = 0.614$). In the multivariate analysis, only younger age was statistically significantly associated with the occurrence of PSA failure ($p = 0.008$). The ROC curve analysis calculated the area under the ROC curve (AUC) as 0.844 and determined a cut-off value of 65 years, at which the sensitivity and specificity were calculated as 77.9 and 87.5%, respectively (Fig 2b).

**Table 2. Predictive significance of clinical factors for the occurrence of PSA bounce.**

| | Univariate | | | Multivariate | | |
|---|---|---|---|---|---|---|
| | HR | (95% CI) | *p*-value | HR | (95% CI) | *p*-value |
| Age | 0.92 | (0.88–0.96) | < **0.001** | 0.92 | (0.88–0.96) | < **0.001** |
| T stage | 0.55 | (0.34–0.89) | **0.015** | 0.60 | (0.37–0.99) | **0.049** |
| Gleason score | 0.69 | (0.17–2.86) | 0.605 | - | | |
| initial PSA | 0.97 | (0.88–1.07) | 0.587 | - | | |
| Prostate volume | 1.02 | (0.99–1.04) | 0.088 | 1.01 | (0.99–1.03) | 0.159 |
| ADT duration | 1.00 | (0.98–1.02) | 0.731 | - | | |
| pre-CIRT PSA | 1.28 | (0.89–1.84) | 0.181 | - | | |
| PSA nadir | 2.55 | (1.10–5.88) | **0.029** | 2.00 | (0.87–4.64) | 0.104 |
| Time to PSA nadir | 0.93 | (0.81–1.06) | 0.284 | - | | |

PSA: Prostate specific antigen, ADT: Androgen deprivation therapy, CIRT: Carbon ion radiotherapy, HR: Hazard ratios, CI: Confidence interval.

Prognostic factors, for which *p* value was calculated as < 0.10, were evaluated by multivariate analysis.

## Discussion

We investigated the dynamics of PSA in patients with prostate cancer who were treated with CIRT and neoadjuvant ADT in the present study. The occurrence of PSA bounce and failure was correlated with younger age. To the best of our knowledge, this is the first report of PSA dynamics after CIRT with neoadjuvant ADT.

Multiple definitions of the PSA bounce have been reported, and no consensus has been established. Several studies used the definition of an increase of >0.2 ng/ml in PSA levels followed by a spontaneous decrease to the pre-bounce level or lower [13, 16, 20, 30, 31]. In the present study, no patient met this definition, as the nadir PSA level was extremely low because of the use of neoadjuvant ADT. Thus, we defined the PSA bounce as an increase of at least 0.4 ng/ml followed by any decrease, in line with previous studies [25, 26]. In a study of PSA bounces in patients treated with conventional external radiotherapy, the bounce was defined

(a)  (b)

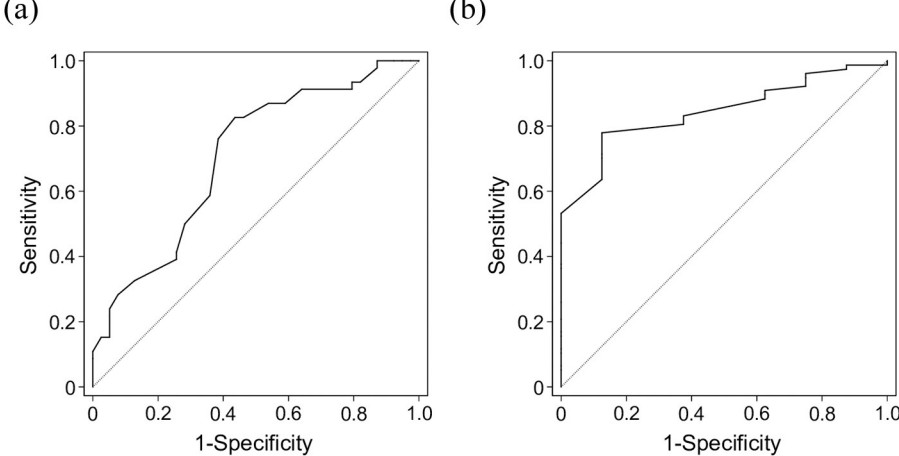

**Fig 2.** (a). ROC curve for the correlation between PSA bounce and age. The area under the ROC curve was 0.705. The cut-off value was 68 years, at which the sensitivity and specificity were 76.1 and 61.5%, respectively. (b). ROC curve for the correlation between PSA bounce and age. The area under the ROC curve was 0.844. The cut-off value was 65 years, at which the sensitivity and specificity were 77.9 and 87.5%, respectively.

**Table 3. Characteristics of patients with PSA failure.**

| Age | T stage | PSA (ng/ml) | Gleason score | Time to failure (month) | PSA dynamics after PSA failure | Time to decreasing PSA after PSA failure (month) |
|---|---|---|---|---|---|---|
| 61 | T2b | 17.09 | 7 | 33 | decreased spontaneously | 3 |
| 61 | T2a | 9.2 | 7 | 21 | decreased spontaneously | 3 |
| 64 | T1c | 11.21 | 7 | 21 | decreased spontaneously | 15 |
| 58 | T2a | 8.43 | 7 | 15 | decreased spontaneously | 3 |
| 54 | T1c | 4.83 | 7 | 15 | decreased spontaneously | 6 |
| 63 | T2a | 14.79 | 7 | 30 | decreased spontaneously | 3 |
| 50 | T2a | 6.27 | 7 | 21 | decreased spontaneously | 3 |
| 68 | T2a | 8.35 | 7 | 24 | decreased by ADT | 3 |

PSA: Prostate specific antigen, ADT: Androgen deprivation therapy.

as an increase of 0.5 ng/ml [32]. Conversely, the PSA bounce was defined as an increase of 0.1 ng/ml followed by two consecutive decreases after IMRT [33].

PSA bounces have been mainly reported after brachytherapy. PSA bounces were observed in 28%–49% of patients after LDR-BT [14]. In two other studies, PSA bounces were observed in 43 and 48% of patients treated with HDR-BT, respectively [34, 35]. In a study of PSA dynamics after HDR-BT combined with conventional external beam radiotherapy, PSA bounces were detected in 31% patients [36].

PSA bounces have also been observed in patients treated with external beam radiotherapy alone. After IMRT, the occurrence rate of PSA bounces ranged 11%–32% [25, 33, 37, 38]. Recently, SRT has been performed for the definitive treatment of prostate cancer, and PSA bounces were also observed after SRT. In a multi-institutional analysis of PSA dynamics, PSA bounces were noted in 26% of patients [18]. Only one study of PSA dynamics after particle beam radiotherapy has been reported [20]. In that study, PSA bounces were observed in 55.7% of patients treated with CIRT alone for prostate cancer.

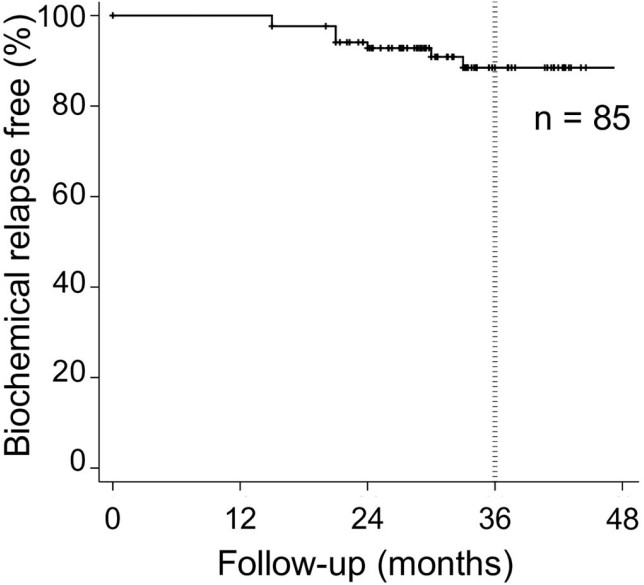

**Fig 3. Prostate-specific antigen (PSA) failure-free rate.** The 3-year PSA failure-free rate was 88.5%.

**Table 4. Predictive significance of clinical factors for the occurrence of PSA failure.**

| | Univariate | | | Multivariate | | |
|---|---|---|---|---|---|---|
| | HR | (95% CI) | *p*-value | OR | (95% CI) | *p*-value |
| Age | 0.86 | (0.79–0.95) | **0.002** | 0.85 | (0.75–0.96) | **0.008** |
| T stage | 1.02 | (0.37–2.82) | 0.962 | - | | |
| Gleason score | NA | - | - | - | | |
| initial PSA | 1.12 | (0.95–1.31) | 0.166 | - | | |
| Prostate volume | 1.03 | (0.99–1.07) | 0.093 | 1,04 | (0.97–1.09) | 0.153 |
| ADT duration | 0.93 | (0.71–1.22) | 0.614 | - | | |
| pre-CIRT PSA | 2.45 | (1.28–4.70) | **0.007** | 1.61 | (0.72–3.57) | 0.244 |
| PSA nadir | 6.67 | (1.88–23.64) | **0.003** | 3.34 | (0.92–12.08) | 0.066 |
| Time to PSA nadir | 0.84 | (0.54–1.31) | 0.443 | - | | |
| PSA bounce | 7.84 | (0.96–63.85) | 0.054 | 2.52 | (0.28–22.93) | 0.411 |

PSA: Prostate specific antigen, ADT: Androgen deprivation therapy, CIRT: Carbon ion radiotherapy, HR: Hazard ratios, CI: Confidence interval, NA: Not available.

Prognostic factors, for which *p* value was calculated as < 0.10, were evaluated by multivariate analysis.

Age was one of the first and most frequently described predictive factors for PSA bounces after brachytherapy [14]. Age was a significant consistent predictor of PSA bounces after IMRT [25, 33]. Regarding other treatment modalities, namely, SRT or HDR-BT combined with external beam radiotherapy, younger age was a significant predictor for PSA bounces [18, 36]. In addition, age was detected as a predictive factor for PSA bounces after CIRT [20]. Similar results were observed in the present study, as younger age was a predictive factor for PSA bounces and PSA failure. Therefore, it is suggested that age is a predictor for PSA bounce regardless of the radiotherapy modality.

In the present study, lower T stage was significantly correlated with PSA bounce. In a study of the PSA bounce after LDR-BT, lower T stage was one of the predictive factors for PSA bounce [39]. Sengoz et al. reported that PSA bounce was more frequent in patients with T1–2 stage cancers after external body radiation therapy [40]. The mechanism by which lower T stage tended to correlate with PSA bounces was unclear. Several previous studies have suggested that T stage has no correlation with PSA bounces [14, 19, 36]. Further investigation is warranted to reveal the correlation between T stage and PSA bounces.

Despite the accumulation of data on post-radiotherapy PSA dynamics, its relevance to clinical outcomes remains unclear. One study suggested that PSA bounces did not predict biochemical recurrence or clinical disease recurrence [33]. Another study reported that PSA bounces after external beam radiotherapy were correlated with PSA failure [26]. By contrast, some reports stated that the PSA bounce was a good predictive factor for PSA failure [19]. Hinnen et al. found that PSA bounces after LDR-BT were predictive of better outcomes [41]. A long-term analysis suggested that the PSA bounce was a significant factor for better overall survival [42]. In CIRT, PSA bounce positivity was a significant predictor of favorable 5-year PSA failure-free survival [20]. In the present study, a significant correlation between PSA bounces and PSA failure was not observed; however, PSA bounces tended to correlate with PSA failure. Longer follow-up is warranted to further explain this issue.

Some patients who exhibited PSA bounces experienced increases in PSA levels of 2 ng/ml or more, which met the Phoenix criteria. The PSA bounce exceeds the 2 ng/ml limit in approximately 10% of patients after brachytherapy [13]. Approximately 1% of patients treated with SRT experienced a PSA increase of >2 ng/ml above the nadir [18]. However, PSA levels spontaneously decreased without any treatment in those patients. A similar clinical course was

observed in the present study, as most patients experienced spontaneous decreases of PSA levels. Therefore, even among patients with PSA increases exceeding 2 ng/ml, which met the PSA failure criteria, continuous close PSA surveillance should be considered to confirm the PSA bounce without immediate treatment such as ADT. These findings may provide important information for both patients and physicians to understand PSA dynamics after CIRT.

The present study had several limitations, such as its single-institutional nature, small number of patients, short observation period, and lack of cases of clinical recurrence. In particular, the small number of PSA failure cases could reduce the power of the statistical analysis. Although the correlation between PSA bounces and androgen production in younger age patients was suggested [43], serum androgen levels were not measured in the present study.

## Conclusions

We observed the dynamics of PSA in patients with prostate cancer who were treated with CIRT and neoadjuvant ADT in the present study. PSA levels should be examined after treatment to survey for clinical recurrence. Further follow-up is needed to reveal the clinical significance of PSA dynamics.

## Author Contributions

**Conceptualization:** Yosuke Takakusagi, Takahiro Oike, Hiroyuki Katoh.

**Data curation:** Yosuke Takakusagi, Kio Kano, Wataru Anno, Keisuke Tsuchida.

**Formal analysis:** Kio Kano, Keisuke Tsuchida, Nobutaka Mizoguchi.

**Investigation:** Yosuke Takakusagi, Kio Kano, Wataru Anno, Keisuke Tsuchida.

**Methodology:** Yosuke Takakusagi, Nobutaka Mizoguchi.

**Project administration:** Yosuke Takakusagi.

**Supervision:** Takahiro Oike, Itsuko Serizawa, Daisaku Yoshida, Hiroyuki Katoh, Tadashi Kamada.

**Validation:** Takahiro Oike, Nobutaka Mizoguchi, Itsuko Serizawa, Daisaku Yoshida.

**Writing – original draft:** Yosuke Takakusagi.

**Writing – review & editing:** Takahiro Oike, Daisaku Yoshida, Hiroyuki Katoh, Tadashi Kamada.

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
