## [Decision Letter · Decision Letter 0]

14 Jul 2020

PONE-D-20-18791

Prostate-Specific Antigen Dynamics After Carbon Ion Radiotherapy for Prostate Cancer

PLOS ONE

Dear Dr. Takakusagi,

Thank you for submitting your manuscript to PLOS ONE. After careful consideration, we feel that it has merit but does not fully meet PLOS ONE’s publication criteria as it currently stands. Therefore, we invite you to submit a revised version of the manuscript that addresses the points raised during the review process.

We look forward to receiving your revised manuscript.

Kind regards,

Stephen Chun

Academic Editor

PLOS ONE

Journal Requirements:

2. In the ethics statement in the manuscript and in the online submission form, please provide additional information about the patient records used in your retrospective study, including: a) the date range (month and year) during which patients' medical records were accessed and b) the source of the medical records analyzed in this work (e.g. hospital, institution or medical center name). If patients provided informed written consent to have data from their medical records used in research, please include this information."

3. We noticed minor instances of text overlap with the following previous publication(s), which need to be addressed:

(1) https://ro-journal.biomedcentral.com/articles/10.1186/s13014-020-01575-7

The text that needs to be addressed involves the first paragraph of the Introduction and the first paragraph of the Results section.

In your revision please ensure you cite all your sources (including your own works), and quote or rephrase any duplicated text outside the methods section. Further consideration is dependent on these concerns being addressed.

"The funders had no role in study design, data collection and analysis, decision to

publish, or preparation of the manuscript."

Reviewers' comments:

Reviewer's Responses to Questions

**Comments to the Author**

1. Is the manuscript technically sound, and do the data support the conclusions?

Reviewer #1: No

Reviewer #2: Yes

Reviewer #3: Partly

2. Has the statistical analysis been performed appropriately and rigorously? 

Reviewer #1: No

Reviewer #2: Yes

Reviewer #3: No

3. Have the authors made all data underlying the findings in their manuscript fully available?

Reviewer #1: Yes

Reviewer #2: Yes

Reviewer #3: No

4. Is the manuscript presented in an intelligible fashion and written in standard English?

Reviewer #1: Yes

Reviewer #2: Yes

Reviewer #3: No

5. Review Comments to the Author

Reviewer #1: Dear authors, thank you for submitting this manuscript, which I read with great interest. Please find my comments below, which I hope will be a positive contribution.

Summary:

This manuscript reviews the retrospective analysis of PSA failure and bounce for the patients with intermediate-risk prostate cancer treated with CIRT combined with neoadjuvant hormonal therapy at a single institution. There is limited information about PSA kinetics about CIRT combined with hormonal therapy.

This reviewer has some significant concerns that need to be addressed by the authors:

Comments:

Abstraction

Line 53, 54:

It would be helpful to have a more specific conclusion based on the findings of the data reported in the paper. Statements with "investigated" are not conclusions.

Background

Line 74-77:The facility's description is considered to be not directly relevant in the background. I recommend that you consider moving it to the methods paragraph.

Materials and Methods

Line 134-135: Describe the details of dose constraints other than the rectum.

Line 156: Statistical Analysis Paragraph

"The correlation of clinical variables with PSA dynamics was assessed via logistic regression." In Table 2 and 4, was logistic regression analysis performed for both univariate and multivariate analyses? If not, please add a different test method used for univariate analysis.

PSA failure and PSA bounce are both time series data and it is not appropriate to use logistic regression in the test method

In the multivariable analysis, the number of independent variables greatly exceeds the number of occurrence events, which lead to inappropriate results. Describe the process of selecting explanatory variables.

Clarify the detail about the statistical methods for differences between groups in the PSA time series used in Figures 1b and 1c.

Results

Tables:

Remove unnecessary gray lines in each table.

Table1: The sum of pretreatment PSA is not 85. Correct the number of patient numbers in each category.

Table2: The results of PSA nadir is not correctly displayed.

Reviewer #2: Given the fact that the clinical significance of PSA bounce is unclear and its definition is various, I think such a prospective observational study is valuable. This manuscript is very important as the first report of PSA dynamics after CIRT with neoadjuvant ADT.

I have one question that there were 8 patients with PSA failure, 7 were followed up, and only 1 was treated immediately. What is the reason for this difference?

Reviewer #3: General comments

Carbon ion radiotherapy requires advanced technique and the patient number is limited compared with those who receive photon radiotherapy. The current study analyzed intermediate risk prostate cancer patients who received CIRT and ADT to illustrate the dynamics of PSA after treatment.

It is essential to further describe the intention to investigate PSA dynamics for prostate cancer after CIRT and ADT and potential clinical impact to emphasize the value of this study.

Specific comments

Materials and methods section

1. How many patients were in the favorable intermediate risk group among the 85 patients? If any, please describe the indication of neoadjuvant ADT use.

2. Please describe the ADT regimen (drug name and dosage) specifically and clarify the last dose of ADT and its active duration.

3. Please explain why patients received neoadjuvant ADT of various duration (4 to 8 months) in this study. Did the authors investigate whether the duration of ADT administration was associated with the time interval of subsequent incidents of PSA bounce/PSA failure?

Statistical analysis section

4. Please describe the starting date to calculate the follow-up time.

5. The current study investigated the correlation between different continuous variables and PSA bounce/PSA failure. ROC curve analysis should be adopted to survey the potential cutoff value of these continuous variables.

6. Cox regression should be considered in multivariate analysis to investigate the effect of these variables on the time it takes for PSA bounce or PSA failure to happen.

Results section

7. The median follow-up time was 33.1 (range, 20.1–48.3) months, which is relatively short for intermediate risk group prostate cancer patients.

8. Compared with conventional fractionated RT for prostate (about 7 to 8 weeks), the treatment course of 3 weeks using CIRT was relatively short. Usually, the decline of PSA after RT is gradual and it usually take several months to reach the PSA nadir. Most patients in the current study reached nadir rapidly within 3 to 4 months after CIRT. How can the authors confirm that the nadir is due to the effect of CIRT (unique RBE?) rather than effect of ADT?

9. Only eight patients developed biochemical failure during follow-up of this cohort. The small number could reduce the power of statistical analysis.

10. Seven of eight patients developed PSA failure and five of them had PSA level decrease spontaneously. Please explain the findings.

11. Only one out of eight patients with PSA failure received salvage ADT. What about the other patients? Did those patients receive any other salvage therapy? Otherwise, the authors should provide the subsequent management and clinical course of those patients.

Discussion section

12. The authors could try to review the published literature to compare the PSA dynamics of intermediate risk prostate cancer patients receiving CIRT, CIRT with ADT, IMRT, and IMRT with ADT.

6. PLOS authors have the option to publish the peer review history of their article (what does this mean?). If published, this will include your full peer review and any attached files.

Reviewer #1: No

Reviewer #2: No

Reviewer #3: No

---

## [Author Response · Author response to Decision Letter 0]

10 Aug 2020

Reviewer #1: 

Thank you for your review and precise suggestions. We have revised our manuscript according to your suggestions as follows: 

“Dear authors, thank you for submitting this manuscript, which I read with great interest. Please find my comments below, which I hope will be a positive contribution.

Summary:

This manuscript reviews the retrospective analysis of PSA failure and bounce for the patients with intermediate-risk prostate cancer treated with CIRT combined with neoadjuvant hormonal therapy at a single institution. There is limited information about PSA kinetics about CIRT combined with hormonal therapy.

This reviewer has some significant concerns that need to be addressed by the authors:”

Comments:

Abstraction

“Line 53, 54:

It would be helpful to have a more specific conclusion based on the findings of the data reported in the paper. Statements with "investigated" are not conclusions.”

 We have revised the Conclusion of the Abstract. A new sentence has been added to explain our study in the Conclusion. (Page 4, lines 54 to 55).

 “This study revealed the significant predictors of PSA bounce and PSA failure.”

“Background

Line 74-77:The facility's description is considered to be not directly relevant in the background. I recommend that you consider moving it to the methods paragraph.”

 The sentence describing our facility has been moved to the Materials and Methods section. (pages 7–8, lines 112–113). 

“Materials and Methods

Line 134-135: Describe the details of dose constraints other than the rectum.”

 V80% < 10ml for the rectum was the only dose constraint in this study. Other DVH parameters for the rectum were not used for dose constraints. We did not use dose constraints for other normal tissues, such as bladder, urethra, penile bulb, and femoral head. 

“Line 156: Statistical Analysis Paragraph

"The correlation of clinical variables with PSA dynamics was assessed via logistic regression." In Table 2 and 4, was logistic regression analysis performed for both univariate and multivariate analyses? If not, please add a different test method used for univariate analysis.

PSA failure and PSA bounce are both time series data and it is not appropriate to use logistic regression in the test method

In the multivariable analysis, the number of independent variables greatly exceeds the number of occurrence events, which lead to inappropriate results. Describe the process of selecting explanatory variables.”

We have revised the statistical analysis according to the comments. The Cox regression model was newly adopted for both univariate and multivariate analyses. Another reviewer suggested that we analyze the ROC curve. We have revised the manuscript to explain the statistical methods in the Materials and Methods section. (Page 10, lines 161–167).

 “The correlation of clinical variables with PSA dynamics was assessed via Cox regression model. Prognostic factors of which p value was calculated as < 0.10 were evaluated by multivariate stepwise Cox regression model [28]. Comparative analyses for continuous variables such as PSA level and age of the two groups were examined using the Mann–Whitney U test. Receiver operating characteristic (ROC) curves were generated and used to determine the optimal cut-off value.”

The results were modified using the Cox regression model. We have revised Tables 2 and 4 to reflect these changes and have revised the text to explain the new results. (Page 15, lines 203–205; page 16, lines 210–214; page 19, lines 238–240; and pages 19–20, lines 244–246).

“In the univariate analysis, younger age, lower T stage and higher PSA nadir were statistically significantly associated with the occurrence of a PSA bounce (p = 0.000, 0.015 and 0.029, respectively).”

“The median PSA nadir of patients with and without PSA bounces were 0.014 (range, 0-2.183) and 0 (range, 0-0.201) ng/ml, respectively (p = 0.002). In the multivariate analysis, younger age and lower T stage were significantly associated with the occurrence of a PSA bounce (p = 0.000 and 0.049, respectively).”

“In the univariate analysis, younger age, higher pre-CIRT PSA levels, and higher PSA nadir were significantly associated with the occurrence of PSA failure (p = 0.002, 0.007, and 0.003, respectively).”

“The median PSA nadir of patients with and without PSA bounces were 0.014 (range, 0-2.183) and 0 (range, 0-0.201) ng/ml, respectively (p = 0.002).”

The Cox regression model revealed that T stage was also correlated with PSA bounces, so we have added explanations of these results to the Results section. (Page 20, lines 212 to 214) We have also added a discussion about the relationship between T stage and PSA bounces to the Discussion section. (Page 24, lines 294 to 301).

“In the multivariate analysis, younger age and lower T stage were significantly associated with the occurrence of a PSA bounce (p = 0.000 and 0.049, respectively).”

“In the present study, lower T stage was significantly correlated with PSA bounce. In a study of the PSA bounce after LDR-BT, lower T stage was one of the predictive factors for PSA bounce [38]. Sengoz et al. reported that PSA bounce was more frequent in patients with T1-2 stage after external body radiation therapy [39]. The mechanism which lower T stage tended to correlate with PSA bounces was unclear. On the other hand, there were several studies suggested that T stage had no correlation with PSA bounces [14, 19, 35]. Further investigation is warranted to reveal the correlation between T stage and PSA bounces.”

We have added a new figure to show the ROC curves (Fig. 2a and 2b).

“Clarify the detail about the statistical methods for differences between groups in the PSA time series used in Figures 1b and 1c.”

We have revised the manuscript to clarify the statistical methods for PSA in each of the two groups. (Page 11, lines 164 to 166).

 “Comparative analyses for continuous variables such as PSA level and age of the two groups were examined using the Mann–Whitney U test.”

“Results

Tables:

Remove unnecessary gray lines in each table.”

The gray lines are not shown in the PDF file. We have attached the PDF file for the tables.

“Table1: The sum of pretreatment PSA is not 85. Correct the number of patient numbers in each category.

Table2: The results of PSA nadir is not correctly displayed.”

We have revised Tables 1 and 2 accordingly.

 

Reviewer #2: 

Thank you for your review and precise suggestions. We have revised our manuscript according to your suggestions as follows: 

“Given the fact that the clinical significance of PSA bounce is unclear and its definition is various, I think such a prospective observational study is valuable. This manuscript is very important as the first report of PSA dynamics after CIRT with neoadjuvant ADT.

I have one question that there were 8 patients with PSA failure, 7 were followed up, and only 1 was treated immediately. What is the reason for this difference?” 

We had not decided the treatment protocol after PSA failure, and each urologist determined the treatment policy for each case. We have to consider the treatment policy for the patient with PSA failure. As the results of the present study showed that PSA level decreased without treatment in many PSA failure cases, we believe that careful follow-up is important.

 

Reviewer #3: 

Thank you for your review and precise suggestions. We have revised our manuscript according to your suggestions. 

“General comments

Carbon ion radiotherapy requires advanced technique and the patient number is limited compared with those who receive photon radiotherapy. The current study analyzed intermediate risk prostate cancer patients who received CIRT and ADT to illustrate the dynamics of PSA after treatment.”

“It is essential to further describe the intention to investigate PSA dynamics for prostate cancer after CIRT and ADT and potential clinical impact to emphasize the value of this study.”

Specific comments

Materials and methods section

“1. How many patients were in the favorable intermediate risk group among the 85 patients? If any, please describe the indication of neoadjuvant ADT use.”

Thirty patients were found to be in the favorable intermediate risk group after NCCN classification. Because we used the D’Amico classification in this study, intermediate risk was not divided into favorable and unfavorable groups. It is still unclear whether ADT is necessary for patients with favorable intermediate risk of prostate cancer treated by CIRT. This is a subject for future analysis.

“2. Please describe the ADT regimen (drug name and dosage) specifically and clarify the last dose of ADT and its active duration.”

We have explained the ADT regimen in the Materials and Methods section. (Page 10, lines 148–149).

 “We performed a representative ADT using a combination of bicaltamide and leuprorelin acetate.”

“3. Please explain why patients received neoadjuvant ADT of various duration (4 to 8 months) in this study. Did the authors investigate whether the duration of ADT administration was associated with the time interval of subsequent incidents of PSA bounce/PSA failure?”

The ADT duration was determined by the results of previous research. We have added a reference to clarify the previous research. (Page 10, line 147).

The ADT duration was not associated with either PSA bounce or PSA failure (Tables 2 and 4).

Statistical analysis section

“4. Please describe the starting date to calculate the follow-up time.”

We have modified the Materials and Methods section to clarify the starting date in order to calculate the follow-up time. (Page 10, lines 157 to 158).

“5. The current study investigated the correlation between different continuous variables and PSA bounce/PSA failure. ROC curve analysis should be adopted to survey the potential cutoff value of these continuous variables.”

We performed a ROC curve analysis. We have added new sentences to the Materials and Methods section to explain the statistical analysis. (Page 11, lines 166–167).

 “Receiver operating characteristic (ROC) curves were generated and used to determine the optimal cut-off value.”

We have added new figures (Fig. 2a and 2b) and sentences to explain the results of the ROC analyses. (Page 16, lines 214–217 and page 20, lines 247–250).

“The ROC curve analysis calculated the area under the ROC curve (AUC) as 0.705 and determined a cut-off value of 68 years, at which the sensitivity and specificity were measured to be 76.1 and 61.5 %, respectively. (Fig 2(a))” 

“The ROC curve analysis calculated the area under the ROC curve (AUC) as 0.844 and determined a cut-off value of 65 years, at which the sensitivity and specificity were calculated as 77.9 and 87.5 %, respectively. (Fig 2(b))”

“6. Cox regression should be considered in multivariate analysis to investigate the effect of these variables on the time it takes for PSA bounce or PSA failure to happen.”

We have revised the statistical analysis section. Cox regression model was newly adopted for both univariate and multivariate analyses. We have revised the Materials and Methods section to explain the statistical methods. (Page 10, lines 161–167). 

 “The correlation of clinical variables with PSA dynamics was assessed via the Cox regression analyses. Prognostic factors, for which p value was calculated as <0.10, were evaluated using the multivariate stepwise Cox regression model [28]. Comparative analyses for continuous variables, such as PSA level and age of the two groups, were examined using the Mann–Whitney U test. Receiver operating characteristic (ROC) curves were generated and used to determine the optimal cut-off values.”

The results were changed by the Cox regression model. We have revised Tables 2 and 4 and have added sentences to explain these results. (Page 15, lines 203 to 205; page 16, lines 210–214; page 19, lines 238–240; pages 19–20, lines 244–246).

“In the univariate analysis, younger age, lower T stage and higher PSA nadir were statistically significantly associated with the occurrence of a PSA bounce (p = 0.000, 0.015 and 0.029, respectively).”

“The median PSA nadir of patients with and without PSA bounces were 0.014 (range, 0-2.183) and 0 (range, 0-0.201) ng/ml, respectively (p = 0.002). In the multivariate analysis, younger age and lower T stage were significantly associated with the occurrence of a PSA bounce (p = 0.000 and 0.049, respectively).”

“In the univariate analysis, younger age, higher pre-CIRT PSA levels, and higher PSA nadir were significantly associated with the occurrence of PSA failure (p = 0.002, 0.007, and 0.003, respectively).”

“The median PSA nadir of patients with and without PSA bounces were 0.014 (range, 0-2.183) and 0 (range, 0-0.201) ng/ml, respectively (p = 0.002).”

The Cox regression model revealed that T stage was also correlated with PSA bounces, so we have added sentences to explain these results in Results section. (Page 20, lines 212 to 213) and have added a discussion regarding the relationship between T stage and PSA bounces in the Discussion section. (Page 24, lines 293 to 300).

“In the multivariate analysis, younger age and lower T stage were significantly associated with the occurrence of a PSA bounce (p = 0.000 and 0.049, respectively).”

“In the present study, lower T stage was significantly correlated with PSA bounce. In a study of the PSA bounce after LDR-BT, lower T stage was one of the predictive factor for PSA bounce [38]. Sengoz et al. reported that PSA bounce was more frequent in patients with T1-2 stage after external body radiation therapy [39]. The mechanism which lower T stage tended to correlate with PSA bounces was unclear. On the other hand, there were several studies suggested that T stage had no correlation with PSA bounces [14, 19, 35]. Further investigation is warranted to reveal the correlation between T stage and PSA bounces.”

Results section

“7. The median follow-up time was 33.1 (range, 20.1–48.3) months, which is relatively short for intermediate risk group prostate cancer patients.”

We also believe that the short follow-up duration is a limitation in this study. A longer follow-up is warranted to reveal the clinical significance of PSA dynamics.

“8. Compared with conventional fractionated RT for prostate (about 7 to 8 weeks), the treatment course of 3 weeks using CIRT was relatively short. Usually, the decline of PSA after RT is gradual and it usually take several months to reach the PSA nadir. Most patients in the current study reached nadir rapidly within 3 to 4 months after CIRT. How can the authors confirm that the nadir is due to the effect of CIRT (unique RBE?) rather than effect of ADT?”

Darivis et al. reported the relationship between ADT and CIRT alone (ref 20). They did not mention the duration for PSA nadir, but a figure in that study demonstrated that it takes several months or more to reach PSA nadir by CIRT alone. Therefore, it is suggested that the short time to PSA nadir in the present study may be affected ADT.

9. Only eight patients developed biochemical failure during follow-up of this cohort. The small number could reduce the power of statistical analysis.

We also believe that the small number of patients with PSA failure is a limitation of the present study. Further investigation is warranted in the future.

10. Seven of eight patients developed PSA failure and five of them had PSA level decrease spontaneously. Please explain the findings.

Because clinical recurrence was not observed in any of the seven patients, careful surveillance of PSA was selected and PSA level was decreased without any salvage treatment. Several cases that met the PSA failure criteria among patients with PSA bounce have been reported in previous studies (13, 18). A similar clinical course was observed in these seven patients in this study.

11. Only one out of eight patients with PSA failure received salvage ADT. What about the other patients? Did those patients receive any other salvage therapy? Otherwise, the authors should provide the subsequent management and clinical course of those patients.

The seven patients with PSA failure did not receive any salvage therapy. We have added the following sentence to clarify this point: 

“No salvage treatments were performed in these seven patients in the follow-up period.” (Page 18, line 226 to 227).

Discussion section

12. The authors could try to review the published literature to compare the PSA dynamics of intermediate risk prostate cancer patients receiving CIRT, CIRT with ADT, IMRT, and IMRT with ADT.

We could not find any previous literature that investigated PSA dynamics limited to intermediate risk groups. Especially for CIRT, there has been only one study into PSA dynamics (Darwis et al.) (20). Therefore, we believe that the present study has novelty and may provide important information for both patients and physicians to understand PSA dynamics after CIRT.

---

## [Decision Letter · Decision Letter 1]

26 Aug 2020

PONE-D-20-18791R1

Prostate-Specific Antigen Dynamics After Carbon Ion Radiotherapy for Prostate Cancer

PLOS ONE

Dear Dr. Takakusagi,

Thank you for submitting your manuscript to PLOS ONE. After careful consideration, we feel that it has merit but does not fully meet PLOS ONE’s publication criteria as it currently stands. Therefore, we invite you to submit a revised version of the manuscript that addresses the points raised during the review process.

We look forward to receiving your revised manuscript.

Kind regards,

Stephen Chun

Academic Editor

PLOS ONE

Reviewers' comments:

Reviewer's Responses to Questions

**Comments to the Author**

1. If the authors have adequately addressed your comments raised in a previous round of review and you feel that this manuscript is now acceptable for publication, you may indicate that here to bypass the “Comments to the Author” section, enter your conflict of interest statement in the “Confidential to Editor” section, and submit your "Accept" recommendation.

Reviewer #1: All comments have been addressed

Reviewer #2: All comments have been addressed

Reviewer #3: (No Response)

2. Is the manuscript technically sound, and do the data support the conclusions?

Reviewer #1: Yes

Reviewer #2: Yes

Reviewer #3: Partly

3. Has the statistical analysis been performed appropriately and rigorously? 

Reviewer #1: Yes

Reviewer #2: Yes

Reviewer #3: No

4. Have the authors made all data underlying the findings in their manuscript fully available?

Reviewer #1: Yes

Reviewer #2: Yes

Reviewer #3: No

5. Is the manuscript presented in an intelligible fashion and written in standard English?

Reviewer #1: Yes

Reviewer #2: Yes

Reviewer #3: No

6. Review Comments to the Author

Reviewer #1: The revised version is well re-written replying the comments of the reviewers. I feel that the revised manuscript is suitable for publication.

Reviewer #2: Given the fact that the clinical significance of PSA bounce is unclear and its definition is various, I think such a prospective observational study is valuable. This manuscript is very important as the first report of PSA dynamics after CIRT with neoadjuvant ADT. I think this paper is worthy for publication.

Reviewer #3: Thank the authors for revising the manuscript; however, some problems remain in the revised manuscript.

1. The topic of the manuscript is “Prostate-Specific Antigen Dynamics After Carbon Ion Radiotherapy for Prostate Cancer.” If the authors were not able to confirm whether the PSA dynamics was due to the effect of CIRT, ADT or both, the issue of this investigation should be “Prostate-Specific Antigen Dynamics After Neoadjuvant ADT and CIRT for Prostate Cancer.” Precisely, the study demonstrated the prostate-specific antigen dynamics after neoadjuvant androgen-deprivation therapy combined with carbon ion radiotherapy for intermediate risk prostate cancer patients.

2. Table 1 demonstrated that the duration of ADT were enormously diverse with median of 6.2 months (ranging from 2.3 to 116.9 months). Essentially, this arises the concern of its potential effects on the time period for PSA bounce or PSA failure. This issue stayed unexplained in the revised manuscript.

3. Only eight patients developed biochemical failure during follow-up of this cohort. The small number could reduce the power of statistical analysis, which should be estimated in the manuscript.

4. Importantly, the statistical analyses have some fundamental problems even after revision. The results of the univariate analysis and Cox regression model of multivariate analysis provided in Table 2 and Table 3 were insufficient to demonstrate how those parameters were correlated with the time period for PSA bounce or PSA failure to happen. For example, if younger age and lower T stage were not clearly defined in the investigation, how did authors conclude that younger age and lower T stage were associated with the time period for PSA bounce or PSA failure to occur after CIRT?

Because CIRT requires advanced techniques and the patient number is limited compared with those who receive photon radiotherapy, this requires careful statistical analysis to investigate PSA dynamics to confirm its clinical impact.

7. PLOS authors have the option to publish the peer review history of their article (what does this mean?). If published, this will include your full peer review and any attached files.

Reviewer #1: No

Reviewer #2: No

Reviewer #3: No

---

## [Author Response · Author response to Decision Letter 1]

9 Sep 2020

Reviewer #1: The revised version is well re-written replying the comments of the reviewers. I feel that the revised manuscript is suitable for publication.

 We thank the reviewer for evaluating our manuscript and for their encouraging comments.

Reviewer #2: Given the fact that the clinical significance of PSA bounce is unclear and its definition is various, I think such a prospective observational study is valuable. This manuscript is very important as the first report of PSA dynamics after CIRT with neoadjuvant ADT. I think this paper is worthy for publication.

 We appreciate the reviewer’s feedback and are elated to hear this news.

Reviewer #3: Thank the authors for revising the manuscript; however, some problems remain in the revised manuscript.

We thank the reviewer for evaluating our manuscript. According to the suggestions, we have revised the manuscript as follows.

1. The topic of the manuscript is “Prostate-Specific Antigen Dynamics After Carbon Ion Radiotherapy for Prostate Cancer.” If the authors were not able to confirm whether the PSA dynamics was due to the effect of CIRT, ADT or both, the issue of this investigation should be “Prostate-Specific Antigen Dynamics After Neoadjuvant ADT and CIRT for Prostate Cancer.” Precisely, the study demonstrated the prostate-specific antigen dynamics after neoadjuvant androgen-deprivation therapy combined with carbon ion radiotherapy for intermediate risk prostate cancer patients.

We have revised the title to “Prostate-Specific Antigen Dynamics After Neoadjuvant Androgen-Deprivation Therapy and Carbon Ion Radiotherapy for Prostate Cancer,” as per their insight (page 1, lines 2–3).

2. Table 1 demonstrated that the duration of ADT were enormously diverse with median of 6.2 months (ranging from 2.3 to 116.9 months). Essentially, this arises the concern of its potential effects on the time period for PSA bounce or PSA failure. This issue stayed unexplained in the revised manuscript.

Tables 2 and 3 demonstrate that the ADT duration was not associated with the occurrence of PSA bounce and failure. This has been delineated in the Results section as follows: 

ADT duration was not correlated with the occurrence of PSA bounce (p = 0.731) (page 16, lines 219–220).

ADT duration was not correlated with the occurrence of PSA failure (p = 0.614) (page 20, lines 258–259).

3. Only eight patients developed biochemical failure during follow-up of this cohort. The small number could reduce the power of statistical analysis, which should be estimated in the manuscript.

We agree with the reviewer’s comment that this small number of patients has contributed to a reduction of statistical power. We have acknowledged this as a limitation of our study, which is as follows:

In particular, the small number of PSA failure cases could reduce the power of the statistical analysis (page 26, lines 341–342). 

4. Importantly, the statistical analyses have some fundamental problems even after revision. The results of the univariate analysis and Cox regression model of multivariate analysis provided in Table 2 and Table 3 were insufficient to demonstrate how those parameters were correlated with the time period for PSA bounce or PSA failure to happen. For example, if younger age and lower T stage were not clearly defined in the investigation, how did authors conclude that younger age and lower T stage were associated with the time period for PSA bounce or PSA failure to occur after CIRT?

Because CIRT requires advanced techniques and the patient number is limited compared with those who receive photon radiotherapy, this requires careful statistical analysis to investigate PSA dynamics to confirm its clinical impact.

In the present study, we solely aimed to determine the occurrence of PSA bounce, PSA failure, and related clinical factors; elucidation of the time period for PSA bounce or failure to occur was beyond the scope of this study. This has been stated in the manuscript as follows: 

In this study, we identified the significant predictors of the occurrence of PSA bounce and failure. (page 4, lines 55–56).

The occurrence of PSA bounce and failure was correlated with younger age. (page 22, lines 273–274).

---

## [Decision Letter · Decision Letter 2]

12 Oct 2020

PONE-D-20-18791R2

Prostate-Specific Antigen Dynamics After Neoadjuvant Androgen-Deprivation Therapy and Carbon Ion Radiotherapy for Prostate Cancer

PLOS ONE

Dear Dr. Takakusagi,

Thank you for submitting your manuscript to PLOS ONE. After careful consideration, we feel that it has merit but does not fully meet PLOS ONE’s publication criteria as it currently stands. Therefore, we invite you to submit a revised version of the manuscript that addresses the points raised during the review process.

We look forward to receiving your revised manuscript.

Kind regards,

Stephen Chun

Academic Editor

PLOS ONE

Additional Editor Comments (if provided):

After additional statistical review, with minor revisions as suggested by Reviewer #4 to clarify methodology, this manuscript will be acceptable for publication.

Reviewers' comments:

Reviewer's Responses to Questions

**Comments to the Author**

1. If the authors have adequately addressed your comments raised in a previous round of review and you feel that this manuscript is now acceptable for publication, you may indicate that here to bypass the “Comments to the Author” section, enter your conflict of interest statement in the “Confidential to Editor” section, and submit your "Accept" recommendation.

Reviewer #4: All comments have been addressed

2. Is the manuscript technically sound, and do the data support the conclusions?

Reviewer #4: Yes

3. Has the statistical analysis been performed appropriately and rigorously? 

Reviewer #4: Yes

4. Have the authors made all data underlying the findings in their manuscript fully available?

Reviewer #4: Yes

5. Is the manuscript presented in an intelligible fashion and written in standard English?

Reviewer #4: Yes

6. Review Comments to the Author

Reviewer #4: In the statistical analysis section, authors should give details on how the ROC curves were used to generate the optimal cutoffs (was it from univariable model? Or multivariable model?) or list any appropriate reference articles.

7. PLOS authors have the option to publish the peer review history of their article (what does this mean?). If published, this will include your full peer review and any attached files.

Reviewer #4: No

---

## [Author Response · Author response to Decision Letter 2]

16 Oct 2020

Reviewer #4: 

We thank the reviewer for evaluating our manuscript. According to the suggestions, we have revised the manuscript as follows.

“In the statistical analysis section, authors should give details on how the ROC curves were used to generate the optimal cutoffs (was it from univariable model? Or multivariable model?) or list any appropriate reference articles.”

We used Youden index to determine the optimal cut-off value. We have revised the statistical analysis section to clarify how the cut off value was generated (Page 11, lines 172-174) and a new reference have been added (ref number 29)

“Non-parametric receiver operating characteristic (ROC) curves were generated and Youden index (J = max [sensitivity + specificity – 1]) was used to determine the optimal cut-off values [29].”

“29) Conroy AL, Liles WC, Molyneux ME, Rogerson SJ, Kain KC. Performance characteristics of combinations of host biomarkers to identify women with occult placental malaria: a case-control study from Malawi. PLoS One. 2011;6(12)”

---

## [Editor Report · Decision Letter 3]

19 Oct 2020

Prostate-Specific Antigen Dynamics After Neoadjuvant Androgen-Deprivation Therapy and Carbon Ion Radiotherapy for Prostate Cancer

PONE-D-20-18791R3

Dear Dr. Takakusagi,

We’re pleased to inform you that your manuscript has been judged scientifically suitable for publication and will be formally accepted for publication once it meets all outstanding technical requirements.

Kind regards,

Stephen Chun

Academic Editor

PLOS ONE
---

## [Editor Report · Acceptance letter]

28 Oct 2020

PONE-D-20-18791R3 

Prostate-Specific Antigen Dynamics After Neoadjuvant Androgen-Deprivation Therapy and Carbon Ion Radiotherapy for Prostate Cancer 

Dear Dr. Takakusagi:

I'm pleased to inform you that your manuscript has been deemed suitable for publication in PLOS ONE. Congratulations! Your manuscript is now with our production department. 

Kind regards, 

on behalf of

Dr. Stephen Chun 

Academic Editor

PLOS ONE